# A Comprehensive Analysis of E-Health Literacy Research Focuses and Trends

**DOI:** 10.3390/healthcare10010066

**Published:** 2021-12-30

**Authors:** Chen Wang, Xiangyi Wu, Huiying Qi

**Affiliations:** 1Department of Health Informatics and Management, The School of Health Humanities, Peking University, Beijing 100191, China; wangchenparis@bjmu.edu.cn; 2The School of Health Humanities, Peking University, Beijing 100191, China; 18811531265@pku.edu.cn

**Keywords:** E-health literacy, bibliometric, research focuses, thematic evolution, development trend, visualization

## Abstract

Objective: To sort out the research focuses in the field of e-health literacy, analyze its research topics and development trends, and provide a reference for relevant research in this field in the future. Methods: The literature search yielded a total of 431 articles retrieved from the core dataset of Web of Science using the keywords “ehealth literacy”, “E-health literacy” and “electronic health literacy”. A bibliometric analysis was performed by using CiteSpace to explore the development history, hot themes, and trends of future research in the field of e-health literacy. Results: The thematic evolution path in e-health literacy was divided into three stages. The research focuses were inspected from four aspects: evaluation, correlation with health-promotion behaviors, influencing factors, and intervention measures for improvement. Conclusion: E-health literacy research faces challenges such as the development of the connotation of the term, the objectivity of evaluation methods, and the long-term impact of interventions. Future research themes in e-health literacy will include the standardization of evaluation instruments and the individualization of therapeutic strategies.

## 1. Introduction

With the rapid development of Internet technology, an increasing number of people are using networks to communicate and search for information in their lives and work. Because of the abundance of health information resources available on the Internet and the ease with which it can be accessed, people are gradually shifting away from traditional health information sources (such as newspapers, periodicals, and doctors’ offices) and toward the Internet. According to Peterson G et al., people commonly use the Internet to hunt for health and pharmaceutical information, and they use this knowledge to play a more active role in their therapy [1]. Chen established an association between searching for health information online and using that information, as well as an association between online medical help-seeking and utilization of online health information [2].

The Internet has made health information more accessible than ever, but there are concerns about the uneven quality of online health information. Especially after the outbreak of COVID-19, the sources of health information have become diverse and filled with false and misleading information [3]. However, people cannot identify true and false network information, which poses a threat to the public. How to overcome the negative effects of online error information and enable the public to quickly obtain accurate health information through networks and maintain their health is a need for the evolution of the times. It is now recognized that enhancing e-health literacy in the population is an effective way to obtain high-quality, web-based health resources [4], and thus, e-health literacy has become an emerging area of research that is gaining public attention.

Eysenbach first proposed the concept of “e-health” in 2001. He defined e-health as an emerging field at the intersection of medical informatics, public health, and business, referring to health services and information delivered or enhanced through the Internet and related technologies. In a broader sense, the term characterizes not only a technical development but also a state-of-mind, a way of thinking, an attitude, and a commitment for networked, global thinking, to improve health care locally, regionally, and worldwide by using information and communication technology [5].” In 2005, the WHO defined e-health as the use of information and communication technologies (ICT) for health [6]. The concept of e-health is the basis of the concept of e-health literacy.

Norman and Skinner first defined electronic health literacy (e-health literacy) as the ability to search, locate, and evaluate health information from electronic resources to solve health problems. Although many scholars have studied the concept of e-health literacy in the later stage, they have not formed a recognized version. To date, the concept and connotation of e-health literacy proposed by Norman and others are most widely cited. They divided e-health literacy into six core competencies: traditional literacy (basic reading, understanding, communication, and writing skills), health literacy (the ability to acquire, understand, evaluate, and apply health information to make decisions related to maintaining or promoting health), information literacy (the ability to access, evaluate, and use information), media literacy (the ability to select, understand, evaluate, and create information media), scientific literacy (the ability to use scientific methods to understand, evaluate, and explain health-related problems), and computer literacy (the ability to solve problems with computers) [7]. Norman stated that the core competencies that make up e-health competency are unlikely to change, although environmental changes could create new challenges for e-health literacy. However, with the increasing application of science and technology in the medical field, the dynamic development in the e-health field has led to continuous changes in the application and understanding of e-health literacy [8].

In recent years, research on e-health literacy has become the focus of many scholars. For example, Norman et al. designed an electronic health literacy scale [9], and CJ McKinley et al. explored the nature of the relationship between informational social support and components of online health information seeking [10]. Xesfingi et al. assessed the eHealth literacy level of citizens, using the eHealth Literacy Scale (eHEALS) [11], to help researchers quickly understand the overall research status and hot spots. From the perspective of bibliometrics, this paper combs and summarizes the development process, thematic evolution, research hotspots, challenges, and development trends in the field of e-health literacy through cluster analysis and thematic evolution analysis, to provide a reference for relevant exploration and research in the future.

## 2. Materials and Methods

### 2.1. Data Source

According to Bradford’s Law, most of the key research appears in core international journals; thus, Web of Science was taken as the data source in this paper. The search strategy was as follows: search through the keywords “ehealth literacy”, “E-health literacy” and “electronic health literacy”, select the item type “article”, set time range from 1900 to 2021, and finally obtain 431 articles; the retrieval time was 27 May 2021.

### 2.2. Toolkits

To explore the research hotspots and trends of e-health literacy, this study conducted a series of bibliometric analyses on the related literature. Bibliometrics is a special type of quantitative analysis in knowledge fields that examines large amounts of scientific literature as its objects of analysis. It generally also uses various literature analysis software programs for visualization analysis and presentation of results. CiteSpace is a literature information visualization tool developed by the team of Professor Chaomei Chen at Drexel University [12]. It performs data analysis based on almost all the retrieved articles and can alleviate incomplete analysis caused by insufficient knowledge and partial literature coverage. We analyzed the research achievement distribution of a subject, subject development, and research trends, and intuitively displayed the results from the analysis by CiteSpace. First, this paper introduces the knowledge map of the time-and-space analysis, including a time distribution map and a space distribution map of ehealth literacy. Second, an analysis on the articles’ national (regional) collaboration networks, conducted using CiteSpace, is presented. Third, a keyword co-occurrence network and co-citation network analysis is presented, showing the research focus of e-health literacy in detail, including key-word extraction and frequency counting, network construction, and research focus analysis. The last section concludes the paper and suggests issues, challenges, and trends for future research.

## 3. Results

### 3.1. Analysis of Time Distribution

The distribution of the number of research papers on e-health literacy can be seen using the interannual variance in the number of published articles. Figure 1 depicts the time distribution of the research literature on e-health literacy in terms of published papers. The first literature on e-health literacy was published in 2006. This study explained the concept of e-health literacy, provided an e-health literacy model, and demonstrated how to tackle the problem of e-health literacy in clinical and public health practice through a series of clinical cases [7]. There were few studies on e-health literacy until 2010; however, the number of relevant articles on this topic increased steadily from 2011 to 2016. After 2014, the number of articles increased dramatically, denoting that research on e-health literacy has received a great deal of attention. Furthermore, to ensure the timeliness of the research topic, this paper also focuses on research results published up to 27 May 2021. In the figure, only a few months of data are included in 2021, so the total number of articles is reduced.

### 3.2. Analysis of Space Distribution

The national (regional) cooperation map displays the productive sources of e-health literacy research and reveals the core strengths and cooperation of relevant research. In the present research, CiteSpace was used to create national (regional) cooperation networks, as shown in Figure 2. A node represents a country/region, the node size represents the total number of published articles, and the different colors in the node correspond to the number of articles in different time periods. The color bar at the top of the figure represents the change in years. The earliest year is on the left, and the further to the right a color is, the more recent the year. Thus, the years at the centers of the circles are the earliest, and further a layer extends outwards, the newer the year it represents. The edges between nodes represent national collaboration, and the different colors of the edge represent the different times of starting cooperation. It can be seen from the network structure that the United States is at the core and has cooperated with many countries. Moreover, their collaborative research started earlier than most other countries and regions.

Table 1 shows the top ten countries/regions by the number of articles published. The United States topped the list by 156 articles, accounting for more than 36%, indicating its emphasis on research in this field. This was followed by Australia and China, both with 37 articles. More than 85% of the total number of published publications came from the top ten nations and regions, indicating that the majority of research efforts on this topic are centered in these countries and locations. The centrality of a node represents the frequency with which it acts as the shortest bridge between the other two nodes. The more often a node functions as an intermediate, the stronger its centrality. Australia ranks first in terms of centrality, with 0.26, and the United States ranks second, with 0.2. The relevance of a node of a country/region in the international cooperation network is proportional to its centrality. Since 2006, related articles have been published in the United States and Canada. Around 2011, Australia (2011), the Netherlands (2011), and South Korea (2012) began conducting relevant research. Other countries and regions started publishing in 2014, albeit late. The US has an advantage in terms of article production and influence.

### 3.3. Keyword Analysis

The association between keywords and research focuses on a specific academic topic can be investigated by analyzing the co-word network of keywords in the article. CiteSpace developed a co-word network of keywords, which is shown in Figure 3. The node represents a keyword. The larger the nodes, the higher the occurrence frequency of the keyword. The color bar at the top of the figure represents the change in years. The earliest year is on the left, and the further the color is to the right, the more recent the year. The different colors in the node indicate the occurrence frequency of the keyword in different time periods, the year of the center of the circle is the earliest, and the further a layer extends outwards, the newer it is. Centrality refers to being in a central position as the keyword intermediary of articles A and B, or having a keyword connecting several articles and playing a pivotal role. The word is central. The number of lines around the node indicates the centrality. The nodes with high centrality are marked with purple outer rings to represent the importance of their keywords.

To clearly present the hot topics of ehealth literacy research, the keywords with frequencies greater than 20 are shown in Table 2. The important nodes with high frequency and centrality, including ehealth literacy (259 times, 0.1), health literacy (127 times, 0.15), information (63 times, 0.12), digital divide (41 times, 0.12), older adults (40 times, 0.13), etc., are key issues in this research field, which serve as references for the following research topic review.

### 3.4. Analysis of Thematic Evolution

This paper employs CiteSpace to produce a co-word timeline view to examine the knowledge structure and evolution path of research subjects in the field of e-health literacy research, as illustrated in Figure 4. The position of the keyword node on the timeline represents the year when the keyword first appeared, and the keyword cluster label generated by CiteSpace is displayed on the right.

Figure 4 shows the development of keywords in each cluster, to understand the thematic evolution path of this research field. Meanwhile, according to the statistical curve of the published article number (see Figure 1), there was an obvious fluctuation trend in 2011 and 2016. Therefore, combined with the analysis of the above two pieces of information, this paper divides the development of ehealth literacy into three stages.

#### 3.4.1. Emergence (Started in 2006)

Due to the rapid advancement of Internet technology, enabling the general public and consumers to broadcast and access accurate health information via electronic means has become a major topic. E-health literacy emerged as a new research subject after Norman CD presented the notion [7] in 2006 and an e-health literacy measurement scale was constructed [9]. In light of the rapid changes in information dissemination and access methods brought on by the Internet and social media, researchers have begun to investigate the factors that influence people’s search for online health information and access to information perception outcomes [13], and to develop training programs to assist people in obtaining and using high-quality Internet health information [14].

#### 3.4.2. Implementation (Started in 2011)

With the increased usage of the Internet, new health inequities are expected to arise in the context of the growth of digital resources in the health area [15]. As a consequence, researchers have begun to focus on the e-health literacy of certain groups (such as college students, the elderly, chronic illness patients, etc.) as well as the discrepancies in literacy among them. Given the worldwide focus on population ageing strategies, more emphasis has been directed to themes such as the health information-seeking behaviors of elderly people, the means to engage in and benefit from e-health initiatives, and solutions to a variety of hurdles experienced at this stage [16]. Current themes also involve how to bridge the digital gap and increase older e-health literacy in an effective manner.

#### 3.4.3. Development (Started in 2016)

The fast growth of health information on websites and mobile phone applications is matched by the expansion of e-health literacy applications. As shown in Figure 4, researchers have continued with the development of more comprehensive and perfect e-health literacy assessment tools and the verification of the effectiveness of the developed tools, especially the applicability verification of the eHealth Literacy Scale (eHEALS) in a multilingual setting. Individuals, communities, and people’s e-health literacy has been more crucial than ever in the face of global health concerns since the widespread spread of coronavirus [17] and constitutes the focus of the current study.

## 4. Analysis of Research Focus in E-Health Literacy

As seen from keyword clustering and thematic evolution, the research on ehealth literacy involves multiple subjects such as the elderly, college students, and patients with different diseases; analyzes a variety of health behaviors such as self-management, quality of life, and physical activity, as well as the digital divide caused by factors such as age and education; and develops relevant tools, as well as promoting ehealth through technology, social media, etc. Therefore, based on the co-word network of keywords, co-word timeline view, high-frequency keyword statistics, clustering, and thematic evolution path, and combined with the content of classical literature, this paper summarizes the ehealth literacy research into four topics: the evaluation of e-health literacy, the correlation between e-health literacy and health-promotion behaviors, influencing factors of e-health literacy, and interventions to improve e-health literacy.

### 4.1. Research on E-Health Literacy Evaluation

An essential foundation for studying public e-health literacy is a scientific evaluation of e-health literacy among different demographics, which sets the groundwork for comprehending the current situation and devising intervention strategies. As indicated in Table 3, academics have created a variety of assessment tools to assess e-health literacy. There are differences in the relevant assessment models of these tools and their application scenarios, applicable groups, evaluation topics, evaluation dimensions, etc.

The eHEALS scale, developed by Norman, is the first self-assessment tool for evaluating e-health literacy. With a total of eight items, the scale attempts to assess six core competencies of e-health literacy, using a five-point Likert rating system to score each item. The higher the score, the better the e-health literacy [9]. The eHEALS scale is one of the most extensively used instruments to evaluate e-health literacy. Several new e-health literacy scales have been created based on this research. The e-HLS (e-health literacy scale) instrument, constructed by Seckin G, has 19 items, including three dimensions of communication, trust, and action [18]; the Digital Health Literacy Instrument (DHLI) devised by Vaart RVD et al. has 21 self-assessment projects and 7 performance-based items that require respondents to apply e-health literacy to answer objective questions [19]. The eHealth Literacy Assessment Toolkit (eHLA), created by Farnoe A et al., contains four health literacy assessment tools and three digital literacy assessment tools [20].

Additionally, several researchers built assessment tools based on the self-developed concept and framework of e-health literacy. Jean BS and others produced the DHLAT (Digital Health Literacy Assessment Tool), a story-based tool for evaluating adolescent e-health literacy [21]. In the hypothetical environment, students individually answer a series of questions to assist a peer in using the Internet to find information about the disease (type 1 diabetes) with which she has recently been diagnosed. Norgaard O. et al. developed the eHealth Literacy Framework (eHLF), which encompasses individual knowledge and skills, systems, and interactions between individuals and systems [22]. Built on eHLF, Kayser L et al. designed an eHLQ (eHealth Literacy Questionnaire) with 35 items in 7 categories, which adds the two components of personal experience and interaction with systems, providing a broader dimension of e-health literacy [23]. Paige SR et al. proposed the transactional model of eHealth literacy (TMeHL), emphasizing communicative features and focusing on individual abilities to interact and exchange information with others while solving health concerns [24]. They generated the Transactional eHealth Literacy Instrument (TeHLI) to assess perceptual abilities associated with the capacity to comprehend, discuss, evaluate, and utilize online health information [25].

Aside from designing assessment tools, eHEALS is frequently utilized since it can test e-health literacy with a brief questionnaire. However, the scale is built based on the context of Britain and America, and only an English version available, so it has to be tested to see if it is also valid in other linguistic situations. As a consequence, researchers from around the world have translated eHEALS into nearly twenty languages for testing and evaluation, including Dutch [26], Japanese [27], German [28], Portuguese [29], Spanish [30], Turkish [31], Italian [32], Korean [33], Hungarian [34], Serbian [35], Polish [36], Chinese [37], Greek [38], Norwegian [39], Amharic [40], Swedish [41], Arabic [42], and Indonesian [43]. The findings indicated that the translated versions have high internal consistency and credibility.

### 4.2. Research on the Correlation between E-Health Literacy and Health-Promotion Behaviors

Health-promoting behaviors include health responsibility, stress management, exercise behavior, dietary behavior, self-realization, and social support, all of which are positive activities or concepts that are beneficial to preserve or promote health [44]. They can assist individuals in avoiding disease, enhancing health, increasing quality of life, and maintaining excellent physical and mental health. Due to the widespread use of the Internet and mobile devices, most people have access to health-related information on the Internet. Individuals with varying levels of e-health literacy range in their ability to seek, comprehend, evaluate, and use online health information, as well as solve health-related problems. Understanding the importance of e-health literacy on health behaviors will equip professionals with the knowledge to enhance population health intervention, increase e-health literacy, and encourage healthy behaviors. Therefore, academics have begun to focus on the link between e-health literacy and health behaviors. Table 4 shows that the level of e-health literacy is a key factor in improving health behaviors.

Health responsibility refers to paying attention to and being accountable for one’s health. Studies have shown that individuals with greater levels of e-health literacy are linked to regular online searches for health information [45], as well as greater frequency of web-based health-seeking actions [46]. Individuals with better e-health literacy can acquire more accurate health-related information, evaluate the quality of information more properly, have better self-management capacity, connect with healthcare practitioners more effectively, and engage in treatment and nursing decision-making [47,48]. Furthermore, in the case of the COVID-19 pandemic, the higher the level of e-health literacy, the greater the willingness to receive vaccination and the better the compliance with public health guidelines [49].

Stress management is the ability to cope with stress. Mental health benefits significantly from e-health literacy. People with e-health literacy can better analyze and alter their health state, avoid negative feelings such as fear and distrust, and enhance their mental health [10]. Individuals with greater levels of e-health literacy are more equipped to cope with challenges, which means they are less likely to suffer from sleeplessness or psychological anguish [50]. Moreover, since the outbreak of COVID-19, a great amount of incorrect and misleading information has been spreading, causing people to be confused and fearful [51]. Individuals with high e-health literacy can better obtain accurate information and manage negative emotions and symptoms, minimizing the epidemic’s frequent mental health issues (such as depression, sleeplessness, and posttraumatic stress disorder) [52].

Nutrition relates to a person’s eating habits and food choices. Healthy food consumption is adversely connected with e-health literacy, but a balanced diet and regular eating habits are favorably correlated [53]. Individuals with better e-health literacy are more likely to adopt healthy eating behaviors (for example, consuming low-fat meals, low-sugar cereals, vegetables, and fruits) [54] because they can more properly search for and interpret information about healthy eating on the Internet [55].

Exercise refers to the regular undertaking of exercise. E-health literacy can positively predict exercise [54]. Individuals with greater literacy are more likely to exercise frequently and participate in sports [55,56] (for example, exercise at least three times a week [57]). Furthermore, an emerging online fitness culture (including health, exercise, and fitness groups or blogs on various social networking sites) disseminates pertinent health and fitness information through online social interaction to inspire and motivate people to live a healthy life. Users of online fitness who have a high level of e-health literacy may better recognize the beneficial information in a large number of mixed materials and modify their lifestyles through appropriate activities [58].

Self-actualization implies the attitude and expectations of life. The link between e-health literacy and quality of life is clear and favorable [59]. Individuals with a high level of e-health literacy may actively create their internal resources to accomplish spiritual growth, giving them a strong feeling of purpose and optimism for the future [60].

Social support describes closeness and intimacy with others. Individuals with high e-health literacy are more likely to be able to solve interpersonal difficulties and sustain meaningful connections with others [60]. At the same time, they may make greater use of interpersonal resources and achieve better results in social relationships [15].

### 4.3. Research on Influencing Factors of E-Health Literacy

With the rapid development of the Internet and the increase in health information from various online sources, investigating the population’s level of e-health literacy and analyzing its influencing factors can help to formulate intervention measures to improve the population’s e-health literacy. Researchers used questionnaires and interviews to gather and evaluate data, and they discovered that the population’s degree of e-health literacy was influenced by a variety of factors, as shown in Table 5.

First, age is associated with e-health literacy. The younger the age, the greater the level of e-health literacy [61]. In terms of gender, women are more likely than men to seek health information on the Internet [62]; as for education, a higher education level is associated with higher e-health literacy [63]; in terms of the aspect of income, people with lower incomes have lower e-health literacy [64]; and as for residential area, the utility of online medical resources in rural populations is lower than that in urban populations [65]. People with good health perceptions are more likely to have e-health literacy, possibly because they are more inclined to seek medical resources before their health deteriorates [66]; additionally, because medical students are more exposed to medical health information in their courses of study, their electronic literacy level is higher than that of other majors [67].

Second, research indicated that a favorable attitude toward the use of online resources is associated with better levels of e-health literacy [66]. Recognizing the utility of receiving health information via the Internet and the significance of making health decisions utilizing Internet resources is another crucial component correlated to e-health literacy [68].

Finally, the motive is the internal driving force that triggers certain behaviors. Individual health awareness, as one of the reasons, has a direct influence on the use of Internet health resources; for example, those who engage in physical activity seem to be more prone to have e-health literacy [11]. Furthermore, confidence in the use and evaluation of network resources will influence e-health literacy. The amount of information literacy [11], frequency of Internet use [69], and network competency [70] are all essential parts of e-health literacy.

### 4.4. Research on Intervention Measures for Improving E-Health Literacy

With the widespread use of information and communication technology in the medical profession, numerous medical and health institutions and organizations are progressively posting health information on the Internet, and the Internet has become an essential source of high-quality health information. However, Internet resources can only contribute if the public has adequate e-health literacy and avoids low-quality materials that are harmful to health. According to research, the amount of e-health literacy is the best predictor of individual health behavior [71]. As a result, researchers began to focus on the design and implementation of intervention measures to foster and promote e-health literacy. Table 6 shows the most often utilized intervention approaches to increase e-health literacy at the moment.

Participants accessing high-quality professional health information websites, using, querying, and learning credible health information offered by websites, and contacting relevant professionals are examples of interventions employing professional health websites. To avoid poor information on the website disrupting the learning effect of participants, high-quality websites sponsored by the government and hospitals were deployed. The results of a study on the effects of the use of professional health websites by diverse groups of teenagers with epilepsy and their parents [72], patients with heart disease [73], the elderly [74], and informal caregivers [75] revealed that participants’ e-health literacy had improved, and they had a positive attitude about the use of websites to impact their health. Furthermore, in addition to providing trustworthy information, aspects such as easy access, user-friendliness, and simple language [76] contributed to e-health literacy education.

Participants in training programs were guided in the process of searching, examining, and assessing electronic health information. Participating in massive open online courses (MOOCs) [77], viewing instructional films [78,79], reading text and graphic materials [80], and taking associated quizzes [81] were all practices of training, and learning techniques included autonomous learning, collaborative learning [82], and discussion learning [83]. Following the completion of the training project, participants’ e-health literacy, ability to search for health information online, knowledge of network health information, and ability to evaluate network information were greatly enhanced. It can be observed that conducting targeted e-health literacy training programs in the population effectively increases public e-health literacy.

Mobile health care apps can provide appropriate information and interventions based on users’ needs economically and efficiently and promote interaction and communication between app providers and users, allowing users to better understand medical information and monitor and manage their health status. Similarly, wearable medical devices can assist users in understanding and evaluating health information from other sources based on their personal experience by collecting and providing feedback on relevant data and resources, leading to subsequent electronic health behaviors. Competent medical health mobile terminals may provide enough health education, hence increasing the population’s e-health literacy.

Mobile health care apps can provide appropriate information and intervention based on users’ needs, in an economical and efficient manner, and promote interaction and communication between app providers and users, allowing users to better understand medical information and monitor and manage their health status [84,85]. Similarly, wearable medical devices can assist users in understanding and evaluating health information from other sources based on their personal experience by collecting and providing relevant data and resources, leading to subsequent electronic health behaviors [86]. Effective medical health mobile terminals can provide adequate health education, hence enhancing public e-health literacy.

Meanwhile, during the recent coronavirus epidemic, with correct information, disinformation, and changing recommendations blended in a massive amount of materials, there was a tremendous need for instructions on how to identify trustworthy health information among them. As a result, the engagement of people with higher e-health literacy in guiding people with lower e-health literacy, such as college students assisting the elderly [87] and volunteer doctors providing the most up-to-date epidemic-related information to the general population [88], can help improve e-health literacy and narrow the digital divide.

## 5. Limitations

This study examined and summarized relevant studies on the subject of e-health literacy. It serves as a reference for future practice and inquiry in this field. However, there are certain limitations to this paper. Only literature in the WOS core dataset was retrieved and analyzed. In addition, the data sample was not exhaustive. A further study will enlarge the scope of the literature to undertake a more thorough overview.

## 6. Conclusions, Challenges, and Future Trends

### 6.1. Conclusions

This study undertook a bibliometric analysis of e-health literacy research, and the primary work and findings are as follows.

We discovered the research trend in the field of e-health literacy by analyzing the time distributions. The study on e-health literacy has, so far, lasted for 15 years, beginning with the introduction of the concept of e-health literacy and the proposal of an e-health literacy model, and has gained increasing interest from scholars. The amount of literature has expanded dramatically, particularly after 2016, and many study outcomes have been obtained.

The spatial distribution was found by analyzing national collaboration networks. The leading nations in the research of e-health literacy are the United States, Australia, China, and Germany, and the United States has exceeded other countries in terms of quantity and influence. Globally, the proportion of cross-border and cross-regional collaboration in e-health literacy research is expanding, and research professionalism, comprehensiveness, and breadth are constantly improving.

The essential literature and thematic evolution path in the case of internet health literacy research were identified by combining and analyzing the literature co-citation network timeline view. In this paper, e-health literacy research is divided into three stages: emergence, implementation, and development. Each stage’s research material is tied to the social context and technical advancement at the time.

This report highlights the thematic review of e-health literacy research using a keyword co-word network analysis. E-health literacy research trends are diverse. Research on the evaluation of e-health literacy serves as a foundation for related research; the influential factors at the level of e-health literacy and health behavior are studied to support understanding population differences and the significance of improving e-health literacy; improving e-health literacy through intervention measures represents another direction. All of these themes underscore the primary societal concerns of e-health literacy research. The findings of this study can be used as a guide for future practice and research in this subject.

Through the study and analysis discussed above, as well as in the contemporary context of the ongoing development of Internet technology, the obstacles and future potential of e-health literacy were revealed. E-health literacy research has challenges such as the establishment and refinement of the connotation of e-health literacy, the validity of assessment instruments, and the sustainability of intervention effects. Furthermore, future research directions in e-health literacy include the standardization of assessment tools and the customization of intervention measures.

### 6.2. Challenges

In recent times, the rapidly evolving network has expanded access to health information, necessitating the capacity of users to acquire and analyze health information. As a result, in the popular era, e-health literacy has become an essential indicator of ability, which directly influences people’s access to health information, utilization, and making health care decisions via the network. However, with the swift advancement of science and technology, as well as the increasing use of electronic health services by the general public, it is critical to understand how to better benefit from the era of digital health services. Related research on e-health literacy faces numerous challenges.

#### 6.2.1. The Connotation of E-Health Literacy Should Be Enhanced

To date, the concept and connotation of e-health literacy made by Norman CD et al. have been the most widely cited [7]. However, with further research and the development of Web 2.0, the means of sharing health information online are changing, and research on the definition of e-health literacy has begun to highlight the interaction between individual and technical factors [8], as well as the impact of social and environmental factors [89]. Although academics have consistently studied the new connotations of e-health literacy, a recognized version has not been developed; therefore, studies on this idea and connotation need to be completed.

#### 6.2.2. The Objectivity of the Electronic Health Scale Is Insufficient

Zrubka Z et al. referred to the level of e-health literacy as a “self-efficacy-related measure” [34]. Because it was assessed by self-reporting, the evaluation findings depended on professional self-perception knowledge. However, this subjective measurement was not the same as the objective and functional examination of e-health literacy abilities. Individuals frequently overestimate their perceived skills, resulting in a disparity between the exam findings and real e-health literacy. Some researchers also attempted to incorporate objective metrics into assessment tools in the hopes of reducing the over- or underestimation of respondents’ e-health literacy, although the applicability of this work requires further investigation and modification [19].

#### 6.2.3. The Sustainability of the Effects of the Intervention Requires Substantial Investigation

Currently, the intervention study sample size is limited (the number of participants is usually between 30 and 300), and the intervention duration is brief (the intervention mostly lasts for two weeks to three months). Although the intervention measures for a small number of people improved participants’ e-health literacy in a short period and yielded positive results, they could not guarantee the duration of the effect of the training or guidance received by the intervention subjects. Therefore, the sustainability of the intervention effect requires further verification.

### 6.3. Future Trends

Based on the existing themes and the evolution of the issue, this study proposed the following difficulties and future opportunities.

#### 6.3.1. Standardization of E-Health Literacy Assessment Tools

The progress of e-health literacy assessment tools corresponds to the evolution of the concept of e-health literacy. As assessment tools were developed before the widespread use of social media and point-to-point resource sharing, early assessment focused mainly on individual abilities. The assessment material has expanded to include persons, technology, and the relationship between the two because the connotation of e-health literacy has evolved. Subsequent studies should be based on a comprehensive and in-depth description of e-health literacy as well as a unified definition to guide the creation and standardization of assessment tools.

#### 6.3.2. Individualized Interventions on E-Health Literacy to Bridge the Digital Divide

Individuals’ e-health literacy is influenced by a variety of factors, leading to wide disparities. The increased breadth and complexity of Internet use, as well as the upsurge of electronic health knowledge, have resulted in unprecedented inequity in the realm of digital health information. This digital gap must be considered when developing e-health literacy programs. Specific or customized modules should be incorporated with standard intervention programs to increase individual e-health literacy and eliminate health disparities based on diverse groups (particularly those with inadequate education, older age, and lower socioeconomic positions).

#### 6.3.3. Electronic Health Literacy Education

The acceptability of new technology begins with people’s acceptability and understanding of knowledge in a specific field. Both electronic health literacy evaluation tools and intervention measures should be developed to better educate the public on health literacy. The government, schools, and industries are all making efforts for e-health literacy education. Continuous e-health literacy education is needed in this transformation era. Approaches to increasing the participation and motivation of the expected audience should be considered for health literacy education.

## Figures and Tables

**Figure 1 healthcare-10-00066-f001:**
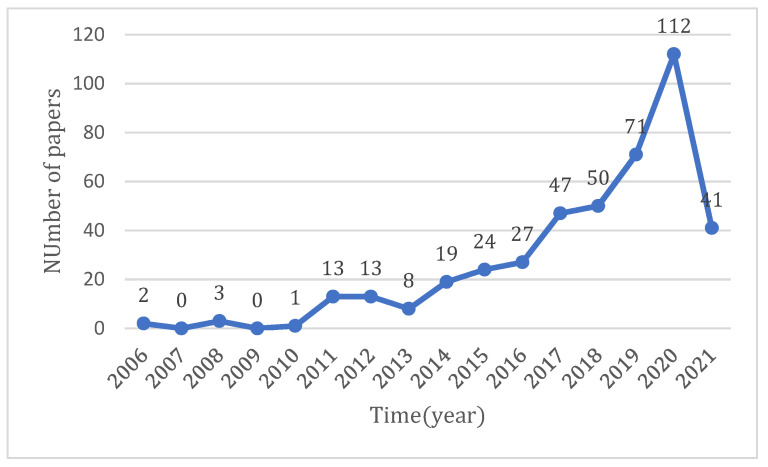
Time distribution graph of research papers on ehealth literacy.

**Figure 2 healthcare-10-00066-f002:**
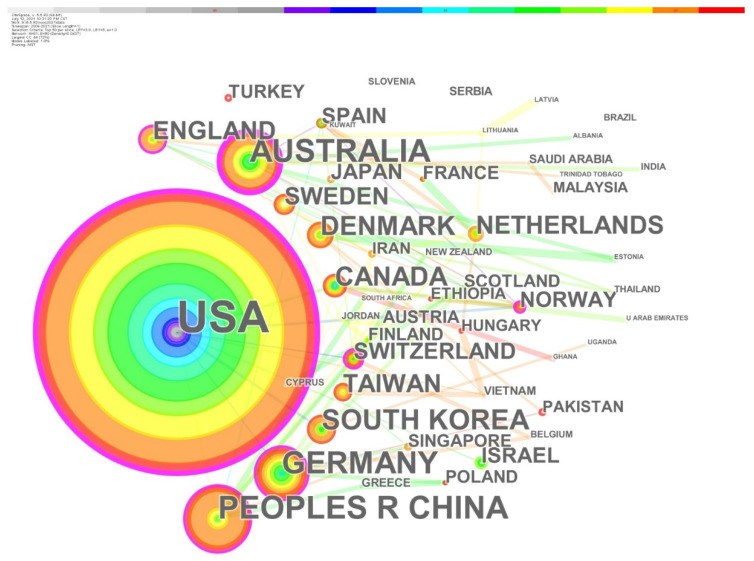
National collaborative network for research on ehealth literacy.

**Figure 3 healthcare-10-00066-f003:**
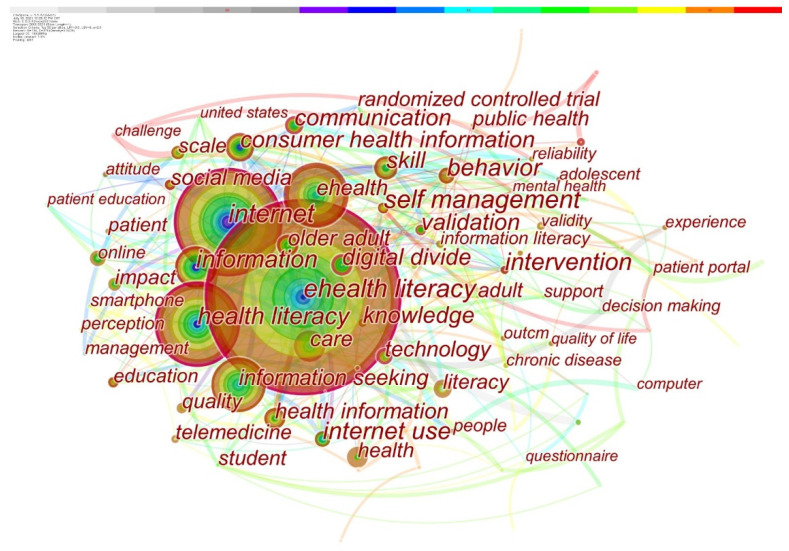
Co-word network of keywords in ehealth literacy research.

**Figure 4 healthcare-10-00066-f004:**
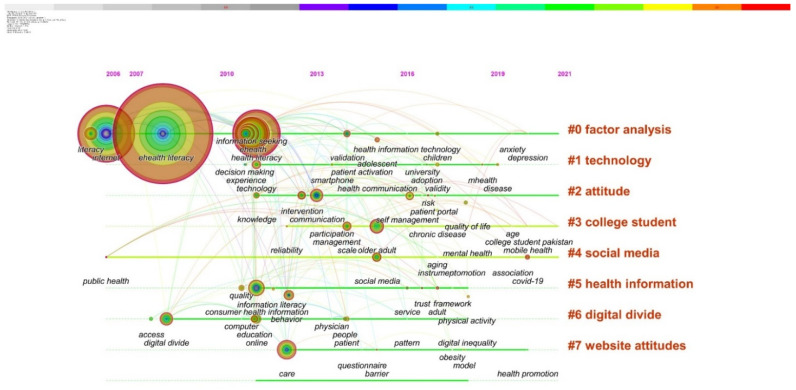
Co-word timeline view of ehealth literacy research literature.

**Table 1 healthcare-10-00066-t001:** Country/region and number of published articles.

Country/Region	No. of Articles	Percentage	Frequency of Citations	Centrality	Year of First Publication
USA	156	36.195%	2987	0.2	2006
Australia	37	8.585%	456	0.26	2011
Peoples R China	37	8.585%	152	0.16	2015
Germany	32	7.425%	390	0.14	2014
South Korea	21	4.872%	226	0.05	2012
Canada	19	4.408%	1119	0.1	2006
Denmark	18	4.176%	160	0.06	2017
England	16	3.712%	202	0.15	2016
Netherlands	16	3.712%	544	0.07	2011
Chinese Taiwan	15	3.480%	188	0.01	2014

**Table 2 healthcare-10-00066-t002:** Keyword frequency and centrality (Frequency > 20).

No.	Keywords	Frequency	Centrality	No.	Keywords	Frequency	Centrality
1	ehealth literacy	259	0.1	15	internet use	37	0.09
2	internet	153	0.1	16	health	36	0.05
3	health literacy	127	0.15	17	behavior	34	0.08
4	ehealth	94	0.05	18	technology	33	0.12
5	information seeking	85	0.05	19	online	31	0.01
6	information	63	0.12	20	scale	30	0.06
7	care	53	0.15	21	impact	30	0.04
8	consumer health information	52	0.08	22	social media	29	0.13
9	skill	47	0.1	23	intervention	28	0.11
10	literacy	42	0.03	24	quality	28	0.03
11	digital divide	41	0.12	25	education	27	0.03
12	health information	41	0.05	26	self-management	26	0.18
13	older adult	40	0.13	27	validation	26	0.14
14	communication	39	0.14	28	knowledge	25	0.09

**Table 3 healthcare-10-00066-t003:** Assessment tools for eHealth literacy.

Name	Content	Dimensions	Literature
eHEALS	8 items	Traditional literacy; Media literacy; Information literacy; Computer literacy; Science literacy; Health literacy	[9]
e-HLS	19 items	Communication; Trust; Action	[18]
DHLI	28 items	Operational skills; Navigation skills; Information searching; Evaluating reliability; Determining relevance; Adding content; Protecting the privacy	[19]
eHLA	96 items	Information need identification and question formulation; Information search; Information assessment; Information management	[20]
DHLAT	13 items	Functional health literacy; Health literacy self-assessment; Familiarity with health and health care; Knowledge of health and disease; Technology familiarity; Technology confidence; Incentives for engaging with technology	[21]
eHLQ	35 items	Using technology to process health information; Understanding of health concepts and language; Ability to actively engage with digital services; Feel safe and in control; Motivated to engage with digital services; Access to digital services that work; Digital services that suit individual needs	[22,23]
TeHLI	18 items	Functional eHealth literacy; Communicative eHealth literacy; Critical eHealth literacy; Translational eHealth literacy	[24,25]

**Table 4 healthcare-10-00066-t004:** Research on the correlation between e-health literacy and health behaviors.

Health-Promotion Behaviors	Conclusions	Literature
Health responsibility	Individuals with better e-health literacy were better able to self-manage and engage in medical decisions, were more willing to be vaccinated, and had greater ability to follow public health guidance.	[45,46,47,48,49]
Stress management	Individuals with better e-health literacy were more likely to be able to control negative emotions and prevent psychological disorders.	[10,50,51,52]
Nutrition	Individuals with better e-health literacy had healthy eating habits and adopted balanced diets.	[53,54,55]
Exercise	Individuals with better e-health literacy levels exercised more frequently with higher participation.	[54,55,56,57,58]
Social support	Individuals with better e-health literacy enjoyed positive interpersonal interactions and are adept at utilizing interpersonal resources.	[59,60]
Self-actualization	Individuals with better e-health literacy had a high quality of life, a sense of purpose, and a sense of hope.	[15,60]

**Table 5 healthcare-10-00066-t005:** Research on influencing factors of e-health literacy.

Factor	Conclusion	Literature
Demographic characteristics	Age, gender, education, income, residential area, health status, and professional differences were associated with e-health literacy levels	[61,67]
Attitude	Attitudes toward accessing Internet resources, as well as an understanding of the Internet’s use and significance, had a substantial influence on the degree of e-health literacy	[66,68]
Motive	Health awareness and confidence in using Internet technology were factors related to e-health literacy	[11,70]

**Table 6 healthcare-10-00066-t006:** Interventions to improve e-health literacy.

Method	Subject	Conclusion	Literature
Professional health website	Teenagers with epilepsy and their parents, heart disease patients, the elderly, informal caregivers, etc.	The content, quality, and feasibility of the site were effective in boosting participants’ electronic health literacy, and participants provided good comments on the intervention, supporting its efficacy and accessibility.	[72,76]
Education video	Patients undergoing coronary angiography, HIV/AIDS patients, Japanese adults, high school students, the elderly, etc.	The development and design of electronic health literacy project training was an effective technique to increase the population’s electronic health literacy, as well as the participants’ self-health management strategies.	[77,83]
Health care mobile terminal	Parents of children with early childhood caries: caries, cancer patients and their caregivers, college students, etc.	To increase users’ electronic health literacy, health APPs and wearable medical devices could give individualized health information and services efficiently.	[84,86]
Consultant	The elderly and the general population under the COVID-19 epidemic	People with low electronic health literacy can benefit from guidance that increases their confidence in accessing Internet technologies and selecting reliable information, which can help narrow the gap.	[87,88]

## Data Availability

Not applicable.

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
