# Peer review of "A Comprehensive Analysis of E-Health Literacy Research Focuses and Trends"

_healthcare, 2021, doi:10.3390/healthcare10010066_

Round 1

Reviewer 1 Report

I would like to thank the authors for addressing such interesting and important topic.

The authors have completed a systematic literature review in the field of e-health literacy.

A bibliometric analysis  of a number of articles was completed by the authors with respect to a number of aspects including research topics and development trends

The introduction section discussed the evolution of seeking information from traditional resources to Internet, which made access to health-related information much easier.

The authors have also discussed challenges when it come to seeking health related information on the Web, especially with challenges related to denitrifying true and false network information

The authors have also introduced and discussed the concept of electronic health (e-health) literacy

The methodology followed by the authors is clearly discussed. Criteria for searching the target databases for relevant articles were presented. The identified articles were analyzed using CiteSpace, Excel for basic statistical analysis, and keyword co-occurrence network and co-citation network analysis.

The authors presented results with respect to the analysis of time distribution of research articles on e-health literacy

CiteSpace analysis showed that the United States is at the core of the publication space and has cooperated with many countries.

The authors presented the analysis of keywords frequencies. The analysis was done using CiteSpace by developing a co-word network of keywords

The authors analyzed themes evolution based on the published articles. These themes are related to emergence, implementation, and development.

Need to revise the paragraph “Authors should discuss the results … may also be highlighted.”

The authors stated that divided e-health literacy research into four categories: the evaluation of electronic literacy, correlation between e-health literacy and health promotion behaviors, the influencing factors of e-health literacy, and measures to improve the e-health literacy of the public

Finally, the authors discussed challenges and future trends in e-health literacy, summarized the results and findings in the conclusion, and discussed limitations of the work.

It is a good idea to provide a short discussion on the significant drop of articles published in 2021 about e-health literacy.

For figure 2, what does the colors refer to? It is a good idea to provide a short discussion on how to interprets the bisexualization in figure 2

For figure 3, it is not clear how the co-word network of keywords was developed. Was it based on the frequencies in the entire documents, the abstract, the title, the keywords, etc..

Figure 4 is not easy to interpret. Please make sure to include a better quality figure.

The authors stated that “e-health literacy research topics are divided into four categories”. However, the authors did not discuss the process based on which the categorization has been done.

What are the criteria for developing tables 2,3,4,5. Have the authors categorized published works on e-health literacy int the four categories?

Need to revise the paragraph “As seen from keyword clustering … and interventions to improve e-health literacy”

Reviewer 2 Report

Dear Editors and Authors:
Thank you for the opportunity to learn about interesting research that is particularly topical today (partly as a result of the pandemic).
I think the paper is solid, interesting and knowledgeable. I therefore broadly recommend its publication. However, there are some aspects that I think could be improved:
1. The introduction is too synthetic. It does not justify the need for the research itself. The concept of e-health is not discussed (only an approximation is given).
2. The objectives and research questions need to be better defined, so that the first part of the article is coherent with the development of the results section.
3. In the methodology, it is necessary to foresee why only Web of Science is used and not the other repositories (SCOPUS, Pubmed). An analysis of the literature that only uses one repository, a priori, leaves many gaps to be studied.
4. In section 2.2 it is necessary to detail the analysis procedure.
5. It is necessary to review the entire referencing process following the journal model. As stated in the journal's own guidelines, "reference numbers should be placed in square brackets [ ], and placed before the punctuation".
6. The conclusions can be improved by taking into account the prospective that the knowledge generated by this research allows (in practical terms).
Good luck to the authors with the review and congratulations on the work.

Reviewer 3 Report

It is a great contribution to the concept of health-literacy, therefore definition analyzes are also important. Are the same concept designations related to the concepts used by different authors? Could it be possible to deepen the current analysis and go further in educational recommendations with the help of the applied tools? It is about indicating what basic knowledge and skills are needed to critically receive information about health, to use information to be able to live in the modern world?Conclusions should not be a summary of the research, but mostly they should formulate some recommendations.
